# Comparative Effect of Low-Glycemic Index versus High-Glycemic Index Breakfasts on Cognitive Function: A Systematic Review and Meta-Analysis

**DOI:** 10.3390/nu11081706

**Published:** 2019-07-24

**Authors:** Celia Álvarez-Bueno, Vicente Martínez-Vizcaíno, Estela Jiménez López, María Eugenia Visier-Alfonso, Andrés Redondo-Tébar, Iván Cavero-Redondo

**Affiliations:** 1Health and Social Research Center, Universidad de Castilla-La Mancha, 16071 Cuenca, Spain; 2Universidad Politécnica y Artística del Paraguay, Asunción 1101, Paraguay; 3Facultad de Ciencias de la Salud, Universidad Autónoma de Chile, Santiago 3467987, Chile; 4Department of Psychiatry, Hospital Virgen de La Luz, 16071 Cuenca, Spain; 5CIBERSAM (Biomedical Research Networking Centre in Mental Health), 28029 Madrid, Spain

**Keywords:** glycemic index, breakfast, cognitive functions, memory, attention

## Abstract

This systematic review and meta-analysis aims to compare the effect of High-Glycemic Index (GI) versus Low-GI breakfasts on cognitive functions, including memory and attention, of children and adolescents. We systematically searched the MEDLINE (via PubMed), EMBASE, Cochrane Central Register of Controlled Trials, Cochrane Database of Systematic Reviews, and Web of Science databases, from their inception until June 2019. Articles comparing the effect of Low-GI versus High-GI breakfasts on the cognitive function (i.e., immediate memory, delayed memory, and attention) of children and adolescents were included. The DerSimonian and Laird method was used to compute the pooled effect sizes (ESs) and their respective 95% confidence intervals (CIs). The pooled ESs were 0.13 (95% CI: −0.11, 0.37) for immediate memory and 0.07 (95% CI: −0.15, 0.28) for delayed memory. For attention, the pooled ES was −0.01 (95% CI: −0.27, 0.26). In summary, GI breakfasts do not affect cognitive domains in children and adolescents.

## 1. Introduction

Breakfast is considered an important part of a balanced diet [1]. Children and adolescents who regularly eat breakfast have a higher intake of micronutrients and tend to have a lower body mass index (BMI) and risk of becoming overweight or obese [1,2,3] than those who do not.

Furthermore, previous studies have shown that breakfast improves cognitive function and academic achievements [4,5,6]. The effect of breakfast on cognitive function has been studied in young adults [7,8,9], and there is an increasing interest in understanding its effect on children’s cognition. While the exact mechanisms of action have not been determined and need to be further studied [10], breakfast increases the level of blood glucose, after a night’s rest, and helps to supply constant levels of blood glucose to the brain, which are needed for regular brain activity and cognitive function.

Glucose is rarely consumed as part of a normal diet; instead, it is obtained from foods containing carbohydrates, which are then broken down into glucose, supplying the organs with the necessary energy [11]. The glycemic index (GI) is a classification of carbohydrate-containing foods, based on the rate of glucose release [12]. GI is estimated by measuring the 2 h postprandial blood glucose concentration, after the consumption of a portion of food containing 50 g of carbohydrates and comparing this estimation with the 2 h postprandial blood glucose concentration, after the consumption of 50 g of glucose [13]. Meals consisting of slowly absorbed carbohydrates would be classified as Low-GI and would cause a slower release of glucose than those meals classified as High-GI or containing quickly absorbed carbohydrates [14].

Some systematic reviews have analyzed the effect of breakfast or GI on cognitive function or academic achievements [15,16,17,18]. These reviews indicate that breakfast consumption has benefits cognitive function to a greater degree than skipping breakfast [15,17,18] and also that a Low-GI meal may favor cognitive function [16]. However, these previous reviews have included different developmental ages (children, adults, and the elderly population) or analyzed the dietary GI [15,16,17,18].

While the effect of consuming breakfast on cognitive function is strongly accepted, the short-term effects of the type of carbohydrates (Low-GI and High-GI) included in a breakfast on the cognitive function of children and adolescents have not been widely studied. Therefore, this systematic review and meta-analysis aims to synthesize the evidence comparing the effect of High-GI versus Low-GI breakfasts on the cognitive functions, including memory and attention, of children and adolescents.

## 2. Materials and Methods

This study is reported in accordance with the Preferred Reporting Items for Systematic Reviews and Meta-Analyses (PRISMA) [19] and follows the recommendations of the Cochrane Handbook for Systematic Reviews of Interventions [20]. This systematic review and meta-analysis has been registered in the PROSPERO data base (registration number: CRD42019136672).

### 2.1. Search Strategy

We systematically searched the MEDLINE (via PubMed), EMBASE, Cochrane Central Register of Controlled Trials, Cochrane Database of Systematic Reviews, and Web of Science databases, from their inception until June 2019. Articles comparing the effect of Low-GI versus High-GI breakfasts on cognitive function (i.e., immediate memory, delayed memory, and attention) in children and adolescents and based on data from experimental studies were eligible. The full search strategy for the MEDLINE database is presented in Table 1. The literature search was complemented by reviewing the citations of the articles considered eligible for the systematic review.

### 2.2. Study Selection

The criteria for the inclusion of studies were as follows: (i) participants—children and/or adolescents; (ii) design—randomized control trials (RCTs), nonrandomized experimental studies (non-RCTs) (including two-arm pre-post studies), and pilot studies; (iii) type of interventions—studies comparing the effect of Low-GI versus High-GI breakfasts on cognitive function; and (iv) outcomes—immediate memory, delayed memory, and attention. The criteria for the exclusion of studies were as follows: (i) reports not written in English or Spanish; (ii) non-eligible publication types, such as review articles, editorials, comments, guidelines or case-reports; (iii) studies combining breakfast interventions with other health interventions, such as physical exercise interventions; and (iv) duplicate reports of the same study.

When more than one study provided data referring to the same sample, we used the study providing more detailed data with the largest sample size. However, data regarding the sample characteristics could also be extracted from multiple reports to obtain the most complete information.

The literature search was independently conducted by two reviewers (C.A.B. and I.C.R.), and disagreements were solved by a consensus or involving a third researcher (V.M.V.).

### 2.3. Data Extraction and the Risk of Bias Assessment

The following data were extracted from the original reports: (1) year of publication; (2) country; (3) study design; (4) sample characteristics (sample size, age distribution, and anthropometric parameters); (5) methods used in the cognition assessment (test and cognitive domains); and (6) type and characteristics of the GI breakfast interventions.

The methodological quality of the RCTs was assessed using the Cochrane Collaboration’s tool for assessing the risk of bias (RoB2) [21]. This tool evaluates the risk of bias according to six domains: randomization, the assignment of intervention, the adherence to the intervention, missing outcome data, and the measurement of the outcome, and selection of the reported results. Each domain was considered to have either a low risk of bias, some concerns relating to the risk of bias, or a high risk of bias. Data extraction and quality assessment were independently performed by two reviewers (C.A.B. and I.C.R.), and inconsistencies were solved by a consensus or involving a third researcher (V.M.V.). The agreement rate between reviewers was reported by calculating kappa statistics.

### 2.4. Statistical Analysis and Data Synthesis

The DerSimonian and Laird method [22] was used to compute a pooled estimate of the effect size (ES) and its respective 95% confidence intervals (CIs) for each of the observed cognitive function domains: immediate memory, delayed memory, and attention. A standardized mean difference score was calculated for each cognitive function domain using Cohen’s d index as the ES statistic [23], in which positive ES values indicate an increase in the cognitive function domain scores, in favor of the Low-GI breakfast intervention versus the High-GI breakfast intervention. Cohen’s d values of around 0.2 were considered to represent weak effects, those around 0.5 were considered to represent moderate effects, those around 0.8 were considered to represent strong effects, and those larger than 1.0 were considered to represent very strong effects [23].

As the included studies report the short-term effects of a GI breakfast on cognitive function domains, post-intervention differences were used to estimate the ES when the studies did not report on pre-intervention cognitive domain values. Additionally, when the studies included two intervention groups or data by gender, their data were analyzed as independent samples. Finally, when the studies performed interventions, combining GI with glycemic load [understood as the combination of the quantity and quality of carbohydrates (GI × amount of carbohydrate/100)] [16], we classified the intervention based on GI.

The heterogeneity of the results across studies was assessed using the I^2^ statistic. I^2^ values were considered as follows: might not be important (0–40%); may represent moderate heterogeneity (30–60%); substantial heterogeneity (50–90%); or considerable heterogeneity (75–100%) [20]. The corresponding *p*-values were also taken into account.

Sensitivity analyses were conducted to assess the robustness of the summary estimates and to detect if any particular study accounted for a large proportion of heterogeneity. Subgroup analyses were performed, based on the age group of participants (children [<12 years] and adolescents [≥12 years]). Random-effects meta-regressions were used to investigate whether the results were associated with the participants’ and intervention characteristics [24] (mean age and BMI), GI values, glycemic load values, and breakfast calories), since these variables could explain the observed heterogeneity.

Finally, the publication bias was evaluated by visual inspection of funnel plots and using the method proposed by Egger [25], considering a *p*-value of <0.10 to be statistically significant [26]. Statistical analyses were performed using STATA SE software, version 15 (StataCorp, College Station, TX, USA).

## 3. Results

### 3.1. Systematic Review

We identified eight studies (Figure 1) addressing the comparative effect of a Low-GI breakfast versus a High-GI breakfast on cognitive function in children and adolescents, which were conducted in three countries: five from the United Kingdom, two from Australia, and one from the United States. The reports were published between 2005 and 2012, and they included studies using the following experimental designs: five were crossover RCTs, one was RCT, one was non-RCT, and one was a pilot study (Table 2) [27,28,29,30,31,32,33,34].

Regarding the characteristics of the populations included, participants were aged between 6 and 15.7 years, with sample sizes ranging from 19 to 74 participants. The mean BMI of the children and adolescents ranged from 17.5 to 22. Only one study did not report on the BMI of the participants [27].

However, the main cognitive function domains measured were immediate memory (including verbal memory, short-term memory, working memory, immediate recall, immediate verbal recall, and verbal recall), delayed memory (including delayed verbal recall, delayed memory, and delayed verbal recall) and attention (including sustained attention, attention switching, accuracy of attention, and immediate attention). Additionally, some studies reported on other cognitive domains (Table 2). The tests were used to assess the cognitive function variations between studies.

Concerning the characteristics of the interventions carried out in the studies, the breakfast GI values used to determine each GI group (Low-GI and High-GI) varied among studies, ranging from 30 to 58 for the Low-GI and from 61 to 314 for the High-GI breakfast. One study classified Oatmeal as Low-GI and Ready-to-eat meals as High-GI. Furthermore, for the Low-GI breakfast interventions, the glycemic load ranged from 5.9 to 43, and the calories ranged from 98 to 502.2. For the High-GI breakfast interventions, the glycemic load ranged from 23 to 55, and the calories ranged from 133 to 468.6.

### 3.2. Risk of Bias

As evaluated by the Cochrane Collaboration’s tool for assessing the risk of bias (RoB2), only one out of eight studies showed a low risk of bias in all domains. In the randomization domain, four out of eight studies showed some concerns relating to the risk of bias, and three out of eight showed a high risk of bias. Only one of the eight studies showed some concerns relating to the risk of bias in adhering to the intervention domain. In the other domains, all the studies showed a low risk of bias (Table 3).

### 3.3. Meta-Analysis

Figure 2 shows the effect of the Low-GI breakfast versus the High-GI breakfast for each cognitive function domain. The pooled ES was 0.13 (95%CI: −0.11, 0.37) for immediate memory and 0.07 (95%CI: −0.15, 0.28) for delayed memory. In the attention domain, the pooled ES was −0.01 (95%CI: −0.27, 0.26). The heterogeneity was substantial for immediate memory (I^2^ = 54.3%; *p* = 0.010) and attention (I^2^ = 55.4%; *p* = 0.022), while it was not important for delayed memory (I^2^ = 7.2%; *p* = 0.373).

#### 3.3.1. Sensitivity Analysis

The pooled ES estimate was not significantly modified when the individual study data were removed from the analysis, one at a time, for any cognitive domain.

#### 3.3.2. Subgroup Analyses and Meta-Regression

Based on the age group of participants, a statistically significant effect on delayed memory was observed for children (ES = 0.33; 95% CI: 0.01, 0.65) (Table 4).

The random-effects meta-regression models showed that the characteristics of the participants [age (*p* = 0.439 for immediate memory, *p* = 0.068 for delayed memory and *p* = 0.620 for attention) and BMI (*p* = 0.607 for immediate memory, *p* = 0.170 for delayed memory and *p* = 0.485 for attention)] were not associated with the pooled ES. Additionally, the characteristics of the intervention (Low-GI and High-GI) [GI values (Low-GI *p* = 0.242 and High-GI *p* = 0.102 for immediate memory, Low-GI *p* = 0.211 and High-GI *p* = 0.159 for delayed memory and Low-GI *p* = 0.572 and High-GI *p* = 0.412 for attention), glycemic load values (Low-GI *p* = 0.123 and High-GI *p* = 0.874 for immediate memory and Low-GI *p* = 0.587 and High-GI *p* = 0.845 for attention) and breakfast calories (Low-GI *p* = 0.719 and High-GI *p* = 0.817 for immediate memory, Low-GI *p* = 0.268 and High-GI *p* = 0.283 for delayed memory and Low-GI *p* = 0.665 and High-GI *p* = 0.638 for attention)] were not associated with the pooled ES (Table 5).

#### 3.3.3. Publication Bias

There was no evidence of a publication bias in both a funnel plot asymmetry and Egger test for immediate memory (*p* = 0.351), delayed memory (*p* = 0.757), and attention (*p* = 0.422).

## 4. Discussion

This systematic review and meta-analysis provide an overview of the evidence concerning the comparative effect of a Low-GI breakfast versus a High-GI breakfast on the cognitive functions, including immediate memory, delayed memory, and attention, of children and adolescents. In summary, GI breakfasts do not affect cognitive domains in children and adolescents except for an effect on delayed memory in children.

Previous systematic reviews have analyzed the effect of breakfast consumption or GI on cognitive function [15,16,17,18] but did not focus on children. While the systematic reviews seem to agree that a Low-GI improves short-term cognitive function, their results were as inconclusive as our findings. However, there is a worldwide consensus that breakfast consumption benefits children’s cognition to a greater degree than morning fasting [18].

Our results showed that the effects of Low-GI vs. High-GI breakfasts were slightly different based on the age group (children or adolescents), especially Low-GI breakfasts, show a positive effect on children’s delayed memory. Some hormonal changes may account for physical, emotional, social, and sexual development during adolescence. These hormonal changes are also responsible for various effects on glucose metabolism, affecting individual glucose tolerance [35,36], which has been suggested as the mechanism to explain the unclear effects of GI on cognitive function. Additionally, children with lower blood glucose levels could show better results in terms of memory, but worse results in terms of attention and time-response, because the blood glucose is absorbed by the brain and tissues [37]. Finally, Low-GI breakfasts provide a slow blood glucose release, which might be especially important in young children, since they have greater brain metabolic demands relative to liver and muscle glycogen and to gluconeogenic capacity [38,39,40,41]. These findings should be cautiously considered because of the small number of studies included in the subgroup analysis, and other factors that could modify the effects of GI on children’s cognition.

When examining the cognition of children and adolescents, we should consider certain factors that may influence cognitive function and therefore influence its relationship with GI. It has been found that girls perform better than boys at school, especially in terms of verbal skills [42]. Additionally, cognition and academic performance have been negatively related to increased adiposity [43,44], and socioeconomic status has been associated with children’s cognitive outcomes [45]. Finally, healthier dietary patterns have been positively associated with attention in adolescence [46]. Unfortunately, most of the studies included in this meta-analysis did not provide information regarding these potential confounding factors. Thus, we could not determine the role of these and other factors in the relationship between GI and cognitive function.

Moreover, cognitive function in children and adolescents could be affected by physical activity in the short and long term, including the physical activity developed in schools, which has an important effect on the cognition and academic performance of children [47,48]. Additionally, the short-term effects of GI on inhibition and working memory could be improved by the combination of a Low-GI breakfast and mid-morning exercise. However, it seems that attention is modified when GI is combined with exercise [49].

Heterogeneity among studies could be considered as substantial for immediate memory and attention, and as not important for delayed memory. Furthermore, the subgroup analysis by age groups (children and adolescents) showed substantial heterogeneity for immediate memory and attention, and no heterogeneity for delayed memory. Although the included studies scored as low risk of bias on most of the RoB2 domains, some aspects affecting the heterogeneity might not be neglected, such as the inclusion of different study designs and differences on GI, glycemic load, and calories values. Finally, the wide range of cut-off to determine low and high GI and the use of different tests to assess cognitive function might also be considered.

Some limitations of this review, which could compromise our results, should be stated. First, data extraction was non-blinded, which is a potential source of bias. Second, the interventions were heterogeneous regarding the GI values and breakfast calories, and some of them included glycemic load manipulation as part of the intervention. However, the meta-regression analyses showed that these factors were not associated with the effect of the intervention on the cognitive domains. Additionally, the experimental study design differs among the included studies, and some biases should not be neglected. Third, none of the studies assessed the dietary GI of children and adolescents outside of the interventions; thus, the total dietary GI could not be included in meta-regression analyses. Fourth, most of the studies did not report their findings by gender. Therefore, we could not explore whether gender differences exist. Fifth, the cognitive domains were measured by different cognition tests, and the definitions of immediate memory, delayed memory, and attention differed among the included studies. Therefore, a misclassification bias could potentially affect our ES estimates. Finally, some language biases could not be avoided, since only studies in English and Spanish were included.

## 5. Conclusions

Our meta-analysis allows us to conclude that breakfast, regardless of whether it is a low- or high-GI breakfast, is effective in improving the cognitive function of children and adolescents. While it seems that a Low-GI breakfast improves memory, especially delayed memory in children, our results are inconclusive. Still, it seems necessary to conduct RCTs with a longer follow-up, larger sample sizes, and long-term results, including academic achievement as an outcome, to elucidate the effects of breakfast GI on the cognitive function of children and adolescents.

## Figures and Tables

**Figure 1 nutrients-11-01706-f001:**
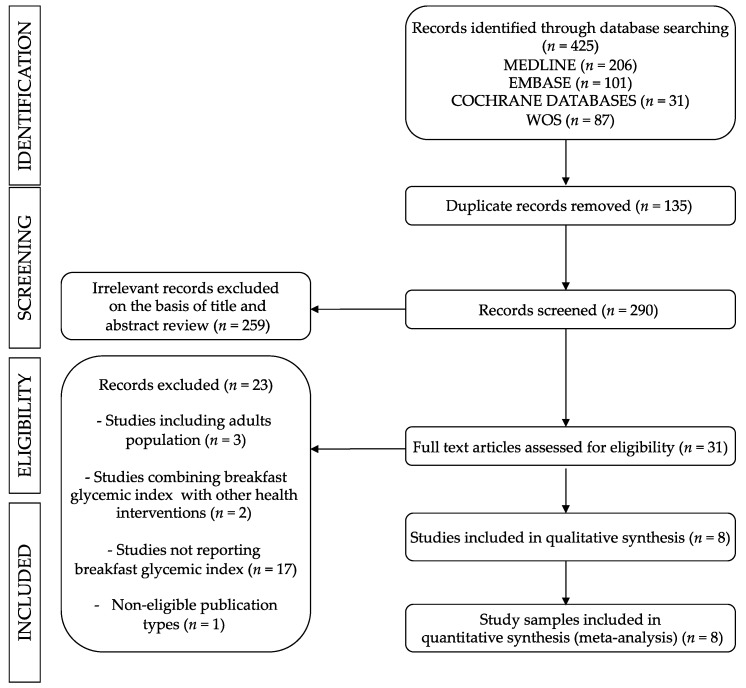
Literature search of the Preferred Reporting Items for Systematic Reviews and Meta-Analyses (PRISMA) diagram.

**Figure 2 nutrients-11-01706-f002:**
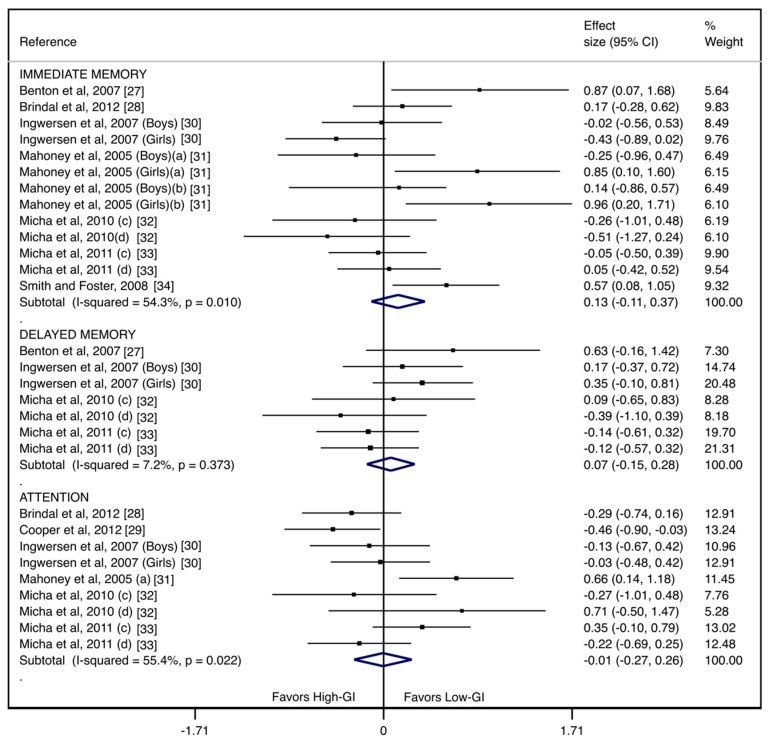
Forest plots of the effect size for the glycemic-index breakfast intervention in children and adolescents, according to the cognitive domains. (**a**) Experiment 1: Mahoney et al. 2005; (**b**) Experiment 2: Mahoney et al. 2005; (**c**) High-glycemic load group; (**d**) Low-glycemic load group. CI: confidence intervals.

**Table 1 nutrients-11-01706-t001:** Search strategy for MEDLINE.

**“Glycemic Index”** **OR** **“Glycemic Index”** **OR** **GI** **OR** **Carbohydrate**	**AND**	**Breakfast** **OR** **Meal** **OR** **“Breakfast Composition”**	**AND**	**Cognition** **OR** **Memory** **OR** **Attention** **OR** **“Cognitive Function”** **OR** **“Cognitive Performance”** **OR** **“Cognitive Processes”**	**AND**	**School** **OR** **children** **OR** **adolescent**

**Table 2 nutrients-11-01706-t002:** Characteristics of the included studies.

Study (Year)	Country	Study Design	Population Characteristics			Outcome		Intervention Characteristics
			Age	Sample Size (n)	BMI (Mean ± SD)	Cognition Test	Cognition Domain	
Benton et al., 2007 [27]	UK	Randomized crossover trial	6.8 (5.9–7.7)	19	NR	(1) Recall of Objects test of the British Ability Scale (2) Paradigm of Shakow	(1) Memory (2) Sustained Attention	- *High GI* GI: 62.8; GL: 39.0; Kcal: 305 - *Medium GI* GI: 50; GL: 5.9; Kcal: 299 - *Low GI* GI: 46; GL: 14.8; Kcal: 284
Brindal et al., 2012 [28]	Australia	Randomized crossover trial	11.6 ± 0.7	39	18.9 ± 3.0	(1) Reaction time response test (2) Attention switching test (3) Finding ‘A’s task (4) Rey Auditory Verbal Learning Test (5) Weschler Intelligence Scale for Children: Digitspan (6) Windows-based program	(1) Speed of processing (2) Attention switching (3) Perceptual speed (4) Short-term memory (5) Working memory (6) Inspection time	- *High GI* GI: 67; GL: 33; Kcal: 314 - *Medium GI* GI: 54; GL: 24; Kcal: 312 - *Low GI* GI: 48; GL: 18; Kcal: 315
Cooper et al., 2012 [29]	UK	Randomized crossover trial	12.8 ± 0.4	41	20.5 ± 3.3	(1) Stroop test (2) Sternberg paradigm (3) Flanker test	(1) Inhibition (2) Working memory (3) Inhibition	- *High GI* GI: 72; GL: 54; Kcal: 422 - *Low GI* GI: 48; GL: 36; Kcal: 420 - *No Breakfast*
Ingwersen et al., 2007 [30]	UK	Randomized crossover trial	9.3 (6.8–11.7)	64	17.5	(1) Cognitive Drug Research (CDR) Computerized Assessment Battery	(1) Speed attention, (2) Speed memory, (3) Accuracy of attention, (4) Secondary memory, (5) Working memory	- *High GI* GI: 77; GL: -; Kcal: 133 - *Low GI* GI: 42; GL: -; Kcal: 98
Mahoney et al., 2005 (a) [31]	USA	Non-RCT	9.0–11.0	30	21.0 ± 6.1	(1) Self-developed spatial map test (2) Digit Span Test (3) Rey Complex Figure Test (RCFT) (4) Continuous Performance Test (CPT) (5) Self-developed prose memory test	(1) Spatial Recall, Learning (2) Immediate Recall, Immediate Attention, Working Memory (3) Visual–Spatial Perception (4) Visual Attention, Vigilance, Auditory Attention (5) Prose Memory	- *Ready-to-eat (high GI)* GI: -; GL: -; Kcal: 200 - *Oatmeal (low GI)* GI: -; GL: -; Kcal: 200 - *No breakfast*
Mahoney et al., 2005 (b) [31]	USA	Non-RCT	6.0–8.0	30	17.7 ± 4.8	(1) Self-developed spatial map test (2) Digit Span Test; (3) Rey Complex Figure Test (RCFT) (4) Continuous Performance Test (CPT) (5) Self-developed prose memory test	(1) Spatial Recall, Learning (2) Immediate Recall, Immediate Attention, Working Memory (3) Visual–Spatial Perception (4) Visual Attention, Vigilance Auditory Attention (5) Prose Memory	- *Ready-to-eat (high GI)* GI: -; GL: -; Kcal: 200 - *Oatmeal (low GI)* GI: -; GL: -; Kcal: 200 - *No breakfast*
Micha et al., 2010 [32]	UK	Pilot study	13.0 ± 0.8	60	20.7 ± 4.6	(1) Word Generation Task (2) Immediate Word Recall (3) Stroop Test (4) Matrices (5) Number Search (6) Serial Sevens (7) Delayed Word Recall	(1) Verbal Fluency (2) Immediate Verbal Recall (3) Alternating Attention, Selective Attention, Impulsivity (4) Visual Reasoning, Nonverbal Intelligence (5) Visual Attention (6) Sustained Attention (7) Delayed Verbal Recall	- *High GL and high GI* GI: 68; GL: 44; Kcal: 378.7 - *Low GL and high GI* GI: 64; GL: 23; Kcal: 240.3 - *High GL and low GI* GI: 53; GL: 43; Kcal: 502.2 - *Low GL and low GI* GI: 58; GL: 31; Kcal: 272.0
Micha et al., 2011 [33]	UK	Randomized crossover trial	12.6 ± 0.9	74	19.3 ± 3.4	(1) Word Generation Task (2) Immediate Word Recall (3) Stroop Test (4) Matrices (5) Number Search (6) Serial Sevens (7) Delayed Word Recall	(1) Verbal Fluency (2) Immediate Verbal Recall (3) Alternating Attention, Selective Attention, Impulsivity (4) Visual Reasoning, Nonverbal Intelligence (5) Visual Attention (6) Sustained Attention (7) Delayed Verbal Recall	- *High GL and high GI* GI: 61; GL: 55; Kcal: 468.6 - *Low GL and high GI* GI: 61; GL: 28; Kcal: 275.6 - *High GL and low GI* GI: 48; GL: 41; Kcal: 469.7 - *Low GL and low GI* GI: 48; GL: 21; Kcal: 281.2
Smith and Foster, 2008 [34]	Australia	RCT	15.7 ± 0.9	38	22.0 ± 3.0	(1) California Verbal Learning	(1) Verbal Recall	- *High GI* GI: 77; GL: -; Kcal: 172 - *Low GI* GI: 30; GL: -; Kcal: 158

BMI: body mass index; GI: glycemic index; GL: glycemic load; RCT: randomized control trial.

**Table 3 nutrients-11-01706-t003:** Risk of bias, assessed by the Cochrane Collaboration’s tool for assessing the risk of bias (RoB2).

	Randomization	Assignment to Intervention	Adhering to Intervention	Missing Outcome Data	Measurement of the Outcome	Selection of the Reported Results
Benton et al., 2007 [27]	Some concerns	Low	Low	Low	Low	Low
Brindal et al., 2012 [28]	Some concerns	Low	Low	Low	Low	Low
Cooper et al., 2012 [29]	Low	Low	Low	Low	Low	Low
Ingwersen et al., 2007 [30]	High	Low	Low	Low	Low	Low
Mahoney et al., 2005 [31]	High	Low	Low	Low	Low	Low
Micha et al., 2010 [32]	High	Low	Some concerns	Low	Low	Low
Micha et al., 2011 [33]	Some concerns	Low	Low	Low	Low	Low
Smith and Foster, 2008 [34]	Some concerns	Low	Low	Low	Low	Low

**Table 4 nutrients-11-01706-t004:** Subgroup Analyses of the Cognitive Function Domains, Based on Age Groups.

	*n*	Effect Size (95%CI)	I^2^
	**IMMEDIATE MEMORY**		
**Children**	8	0.23 (−0.13, 0.59)	62.5
**Adolescents**	5	0.03 (−0.31, 0.36)	45.8
	**DELAYED MEMORY**		
**Children**	3	0.33 (0.01, 0.65)	0.0
**Adolescent**	4	−0.14 (−0.41, 0.14)	0.0
	**ATTENTION**		
**Children**	4	0.04 (−0.36, 0.44)	62.4
**Adolescent**	5	−0.05 (−0.44, 0.35)	58.2

CI: confidence interval. *n* represents the number of studies included for each age group.

**Table 5 nutrients-11-01706-t005:** Random-Effects Meta-Regressions of the Glycemic Index Breakfast Intervention Effect Size by the Characteristics of the Participants and Characteristics of Each Intervention.

	Immediate Memory			Delayed Memory			Attention		
	*n*	ß (95% CI)	*p*	*n*	ß (95% CI)	*p*	*n*	ß (95% CI)	*p*
**Characteristics of the participants**									
Age (years)	13	−0.04 (−0.15, 0.07)	0.439	7	−0.12 (−0.26, 0.01)	0.068	9	−0.05 (−0.27, 0.17)	0.620
BMI (kg/m2)	12	0.04 (−0.15, 0.24)	0.607	6	−0.16 (−0.42, 0.10)	0.170	9	0.08 (−0.18, 0.34)	0.485
**Characteristics of the Low-GI breakfast intervention**									
GI	9	−0.02 (−0.06, 0.02)	0.242	7	−0.03 (−0.09, 0.03)	0.211	8	0.02 (−0.06, 0.09)	0.572
Glycemic load	6	−0.02 (−0.05, 0.01)	0.123	5	NA	NA	6	0.01 (−0.04, 0.06)	0.587
Calories (Kcal)	13	−0.00 (−0.00, 0.00)	0.719	7	−0.00 (−0.00, 0.00)	0.268	9	−0.00 (−0.00, 0.00)	0.665
**Characteristics of the High-GI breakfast intervention**									
GI	9	0.00 (−0.00, 0.01)	0.102	7	0.00 (−0.00, 0.01)	0.159	8	−0.02 (−0.06, 0.03)	0.412
Glycemic load	6	0.00 (−0.04, 0.05)	0.057	5	NA	NA	6	−0.00 (−0.05, 0.04)	0.845
Calories (Kcal)	13	−0.00 (−0.00, 0.00)	0.817	7	−0.00 (−0.00, 0.00)	0.283	9	−0.00 (−0.00, 0.00)	0.630

GI: glycemic index; CI: confidence interval.

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
