# Peer review of "Comparative Effect of Low-Glycemic Index versus High-Glycemic Index Breakfasts on Cognitive Function: A Systematic Review and Meta-Analysis"

_nutrients, 2019, doi:10.3390/nu11081706_

Reviewer 1 Report

Comments and suggestions:

The systemic review and meta-analysis is to compare effect of low versus high glycemic index breakfast on cognitive function among children and adolescents. This is an interesting finding as finds low and high GI breakfast equally effective. However, few minor comments and suggestions can possibly highlight the findings of this analysis to the readers in a better way.

 1.      Figure 1 : is unclear in the version I got.

 2.      Results:

Did you try stratifying by gender in overall? Boys, girls subgroups.

N in table 4 too small.

Confounding factors addressed in each study should also be discussed briefly.

 3.      Discussion:

Include some big cohort studies on dietary pattern and cognition in children and adolescent also. Overall diet quality can also be one important factor affecting cognitive function.  For e.g. Henriksson P et al. Diet quality and attention capacity in European adolescents: the Healthy Lifestyle in Europe by Nutrition in Adolescence (HELENA) study.

 4.      Describe the non-RCT studies design little more and their limitation.

Author Response

REVIEWER #1

Specific comment:

The systemic review and meta-analysis is to compare effect of low versus high glycemic index breakfast on cognitive function among children and adolescents. This is an interesting finding as finds low and high GI breakfast equally effective. However, few minor comments and suggestions can possibly highlight the findings of this analysis to the readers in a better way.

Specific comment:

1.      Figure 1: is unclear in the version I got.

Authors:

Thanks for the reviewer’s comment, we have improved the figure quality.

 Specific comment:

2.      Results:

Did you try stratifying by gender in overall? Boys, girls subgroups.

Authors:

We would like to thank the reviewer’s thoughtful comment. Unfortunately, we could not perform the analysis by gender as there were not enough studies reporting the effects of glycemic index on cognition by sex. We have included a limitation on “limitations” section to state this matter.

“Fourth, most of the studies did not report their findings by gender. Therefore, we could not explore whether gender differences exist.”

 Specific comment:

N in table 4 too small.

Authors:

Thanks for the comment. As the “n” in table 4 represents the number of included studies in each subgroup analysis, we have included a footnote to clarify this statement.

“n represents the number of studies included for each age group.”

 Specific comment:

Confounding factors addressed in each study should also be discussed briefly.

Authors:

Thanks for the reviewer’s comment. We have included a paragraph in “discussion section” to discuss on some of the confounding factor that should be considered when analyzing the effects of glycemic index on cognition.

“When examining the cognition of children and adolescents, we should consider certain factors that may influence cognitive function and therefore influence its relationship with GI. It has been found that girls perform better than boys at school, especially in terms of verbal skills [40]. Additionally, cognition and academic performance have been negatively related with increased adiposity [41,42], and socioeconomic status has been associated with children’s cognitive outcomes [43]. Finally, healthier dietary patterns have been positively associated with attention in adolescence [44]. Unfortunately, most of the studies included in this meta-analysis did not provide information regarding these potential confounding factors. Thus, we could not determine the role of these and other factors in the relationship between GI and cognitive function.”

Specific comment:

3.      Discussion:

Include some big cohort studies on dietary pattern and cognition in children and adolescent also. Overall diet quality can also be one important factor affecting cognitive function. For e.g. Henriksson P et al. Diet quality and attention capacity in European adolescents: the Healthy Lifestyle in Europe by Nutrition in Adolescence (HELENA) study.

Authors:

Thanks for the reviewer’s suggestion. We have included the above-mentioned reference in the manuscript.

“Finally, healthier dietary patterns have been positively associated with attention in adolescence [44].”

Specific comment:

4.      Describe the non-RCT studies design little more and their limitation.

Authors:

Thanks for the reviewer’s comment. We have included in “study selection” section some additional information on the study design.

“(ii) design—randomized control trials (RCTs), nonrandomized experimental studies (non-RCTs) (including two-arm pre-post studies), and pilot studies;”

“Additionally, the experimental study design differs among the included studies, and some biases should not be neglected.”

Reviewer 2 Report

Interesting paper that has practical implications and brings up an important topic for investigation.

I thought the methods were explained well. At some point in the paper, likely when GI is explained, glycemic load needs to be explained as some of the studies used GL, some GI. The influence of using GI vs GL could be discussed.

Unsure why P-value of 0.10 was used rather than 0.05 (page 4).

There are some grammatical issues with the paper that need to be resolved. I made some edits in the introduction, then decided to focus on the paper's content. I strongly suggest that the paper be edited for best wording and grammar.

I enjoyed reading this paper.

Author Response

REVIEWER #2

Interesting paper that has practical implications and brings up an important topic for investigation.

 Specific comment:

I thought the methods were explained well. At some point in the paper, likely when GI is explained, glycemic load needs to be explained as some of the studies used GL, some GI. The influence of using GI vs GL could be discussed.

Authors:

We would like to thank the thoughtful reviewer’s comment. We have written a statement in “statistical analysis and data synthesis” section to clarify that when papers develop interventions combining glycemic index with glycemic load, only the glycemic index classification was considered to determine the intervention group for our meta-analysis.    

“Finally, when the studies performed interventions, combining GI with glycemic load [understood as the combination of the quantity and quality of carbohydrates (GI × amount of carbohydrate /100)] [16], we classified the intervention based on GI.”  

“Second, the interventions were heterogeneous regarding the GI values and breakfast calories, and some of them included glycemic load manipulation as part of the intervention. However, the meta-regression analyses showed that these factors were not associated with the effect of the intervention on the cognitive domains.

Specific comment:

Unsure why P-value of 0.10 was used rather than 0.05 (page 4).

Authors:

Thank you for reviewer’s comment. We have included a reference justifying p-value threshold is 0.10 in Egger test.

26. Egger, M., Smith, G.D., Schneider, M., & Minder C. Bias in meta-analysis detected by a simple, graphical test. BMJ 1997;315:629–34.

Specific comment:

There are some grammatical issues with the paper that need to be resolved. I made some edits in the introduction, then decided to focus on the paper's content. I strongly suggest that the paper be edited for best wording and grammar.

Authors:

We would like to thank the reviewer’s comment. As suggested, English editing has been performed thought the manuscript.

 Specific comment:

I enjoyed reading this paper.

Authors:

Finally, we sincerely appreciate the reviewer’s comment.
